# Fabrication of Micro-Cantilever Sensor Based on Clay Minerals for Humidity Detection

**DOI:** 10.3390/s23156962

**Published:** 2023-08-05

**Authors:** Yiting Xu, Song Liu, Junfeng Zhang, Songyang Chai, Jianjun Li, Changguo Xue, Shangquan Wu

**Affiliations:** 1School of Materials Science and Engineering, Anhui University of Science and Technology, Huainan 232001, China; 2CAS Key Laboratory of Mechanical Behavior and Design of Materials, University of Science and Technology of China, Hefei 230026, China

**Keywords:** humidity sensor, micro-cantilever, montmorillonite, kaolinite

## Abstract

In this paper, novel humidity sensors based on montmorillonite, kaolinite, and composite films coated on micro-cantilevers were prepared to measure the relative humidity (RH) values by the deflection of a micro-cantilever (MC) at room temperature. The humidity-sensing properties, such as response and recovery, sensitivity, repeatability, humidity hysteresis, and long-term stability, were investigated in the range of working humidity (10–80% RH). The humidity response in the close humidity range of 10% RH to 80% RH revealed a linear increase in water absorption of montmorillonite, kaolinite, and montmorillonite/kaolinite mixed dispersant (1:1) as a function of RH with linear correlation factors between the humidity change and deflection estimated to be 0.994, 0.991, and 0.946, respectively. Montmorillonite’s sensitivity was better than kaolinite’s, with the mixed-clay mineral film’s response falling somewhere in between. This research provides a feasible and effective approach to constructing high-performance MC humidity sensors that can be operated at room temperature based on clay minerals.

## 1. Introduction

Humidity sensors are increasingly required in many sectors depending on their detection accuracy, detection range, application conditions, and size scale as the performance requirements for current instruments continue to change. The most prevalent types of humidity sensors are thermal, resistive [1], and capacitive [2]. All three types of humidity sensor can track minute environmental alterations and determine the air’s relative humidity [3,4]. With the increase in the integration of various humidity sensors, miniaturization becomes inevitable in developing sensor systems. Over the past few years, the use of microelectromechanical systems (MEMS)-based sensors in humidity detection has increased tremendously because of the advantages offered by miniaturized sensors over conventional sensors, such as the low cost of manufacturing, high sensitivity, quick response, low power consumption, etc. Among the many MEMS mechanical platforms, micro-cantilevers have become the most popular because of their simple design, fast response-recovery speed [5] ease of fabrication, low cost [6], higher sensitivity [7,8], etc. [9]. In addition, the design of better humidity sensors requires careful consideration of humidity-sensing materials [10].

Functional materials, such as silicon carbide [11], tungsten disulfide [12], silicon dioxide [13], zinc oxide [14], cadmium sulfide [15], vanadium pentoxide [16], and carbon nanotubes [17], are commonly used as humidity-sensing materials. Some synthetic or composite functional materials are expensive, non-biodegradable, or produce environmentally harmful waste products. By comparison, green and non-polluting humidity-sensing materials are gaining popularity due to their ecologically friendly aspects. Clay minerals, commonly found in the natural environment, are used in many industrial applications due to their good properties, such as swelling [18,19,20], adsorption [21,22,23,24], wetting [25,26,27], and catalysis [28,29,30]. The porous structure of clay minerals provides space for water adsorption; therefore, the water adsorption of clay minerals and adulterated minerals has been studied. As common clay minerals, both montmorillonite and kaolinite are naturally occurring and have strong hydrophilic characteristics and water absorption. Feng et al. [31] recorded significantly greater water absorption capacity of montmorillonite when compared to other clay minerals. In addition, kaolinite was more substantial than other clays such as illite. Montmorillonite and kaolinite as natural humidity-sensing materials coated on micro-cantilever (MC) arrays are expected to yield high-performance and integrated humidity sensors.

Herein, novel micro-cantilever humidity sensors were developed based on clay minerals films as humidity-sensing materials. The absorption of water molecules by the clay mineral coating resulted in the cantilever producing an offset reflecting the relative humidity of the surrounding air. The performances of sensors coated by montmorillonite, kaolinite, and their mixture (1:1) in terms of stability, repeatability, wet hysteresis, and associated humidity response sensitivity were thoroughly examined. The experimental findings revealed relevant low-cost, environmentally friendly, and versatile humidity sensors with good measurement reproducibility.

## 2. Experimental

### 2.1. Materials

The kaolinite and montmorillonite were purchased from Pioneer Nano, Nanjing, China. The micro-cantilever (500 µm in length, 90 µm in breadth, and 5 µm in thickness) was obtained from Micromotive, Germany. To facilitate the presentation, “M” and “K” stand for montmorillonite and kaolinite, respectively. The letter “MK” denotes a 1:1 mixture of the two minerals.

### 2.2. Preparation of Humidity Sensors

The process consisted of dispersing 100 mg of montmorillonite powder and 100 mg of kaolinite powder in 4 mL of deionized water to form a montmorillonite solution and a kaolinite solution at concentrations of 25 mg/mL. Following that, 50 mg of montmorillonite powder and 50 mg of kaolinite powder were then dispersed in 4 mL of water by sonication for 6 h to yield a homogeneous montmorillonite/kaolinite composite-clay mineral dispersion (25 mg/mL) with a 1:1 mass ratio of montmorillonite to kaolinite. Trace amounts of prepared montmorillonite, kaolinite, and 1:1 mixed-clay mineral dispersion were taken, and a simple coating method was used in this experiment, where a clay mineral layer with a film thickness of about 1 µm was applied to the sensor under a microscope using the surface tension of the droplets. Three MC humidity sensors (MC-M, MC-K, and MC-MK) were prepared, followed by natural air-drying at room temperature (25 °C) for 24 h.

### 2.3. Apparatus and Characterizations

The optical lever MC relative humidity sensor experimental setup consisted of three parts: (i) wet temperature monitoring, (ii) humidity regulation, and (iii) sensor testing (Figure 1). The MC sensor to be detected is placed in a relatively closed chamber, the gas pipeline is connected through the chamber air inlet, and a commercial hygrothermograph is placed at the air outlet for calibration of the relative humidity and temperature in the chamber. When a laser is directed at the tip of the MC, the light is reflected onto a position-sensitive device (PSD). As the modified clay film will react with expansion or contraction at different relative humidity levels, it causes the MC to deflect, which causes the spot position of the reflected beam on the PSD to change. The length (*l*) of the free end of the MC, and the length (*L*) of the optical path from the laser at the free end of the beam to the PSD can be determined. The PSD measures the deflection changes of the MC and transmits the test data captured by the data acquisition (DAQ) system to the computer data-processing software for processing to complete the testing and recording of the sensor. Using the change (ΔS) of the spot displacement on the PSD, the relationship between the above parameters and the deflection (ΔZ) of the MC can be obtained by arithmetic:(1)$$\Delta Z=\frac{l\times \Delta S}{4L}$$

After placing prepared sensors inside the humidity-controlled chamber, adjusting the humidity of the chamber by controlling the ratio of dry nitrogen (N_2_) and N_2_ flowing through deionised (DI) water, 10% RH was employed as the reference humidity level, and five test humidity points were selected sequentially from the relative humidity range of 10% RH to 80% RH (10% RH, 27% RH, 45% RH, 63% RH, and 80% RH) in the chamber. The experiment was performed at room temperature (25 °C). Temperature and humidity recorders were used to measure the temperature range (−40–80 °C), relative humidity range (0–100% RH), and error (3% RH). The humidity level in the chamber was then raised to the desired level. After achieving the humidity point and a stable sensor, the sensor deflection at the humidity point was recorded.

The morphology features of sensitive films were observed using a scanning electron microscope (SEM, ZEISS GeminiSEM 300). Fourier transform infrared (FTIR) spectroscopy was performed on a NICOLET 380 FTIR spectrometer (Thermo Scientific, Waltham, MA, USA) over a range from 400 cm^−1^ to 4000 cm^−1^. An X-ray diffractometer (XRD, Smartlab SE, Rigaku Corporation, Tokyo, Japan) was used to analyze the crystal structure of clay powders. The samples were tested in a dry state. Powder X-ray diffraction (XRD) experiments were performed between 5° and 80° (2θ).

## 3. Results and Discussion

### 3.1. Characterization of Clay Minerals

The SEM images of the montmorillonite film are presented in Figure 2a, and the porous montmorillonite film structure consisted of several stacked layers conducive to the adsorption of water molecules. The curved pore walls increase the specific surface area of montmorillonite. Kaolinite clay pores looked more developed, with a high void ratio and irregular pore structure, which were conducive to water molecule adsorption (Figure 2b). Meanwhile, the kaolinite film showed primarily a hexagonal shape mixed with montmorillonite in the pore channels and lamellae (Figure 2c). Compared to the kaolinite membranes (Figure 2e), the surface morphology of the montmorillonite membranes (Figure 2d) looked more irregular despite improvement by mixing with kaolinite (Figure 2f). The smoother membrane surface of kaolinite agreed well with its lower specific surface area.

Figure 3 shows the XRD patterns of pure montmorillonite, pure kaolinite, and their mixture. It is clear from the graphs that kaolinite, montmorillonite, and quartz are the main components of the mixed-clay minerals (MK), and no new minerals are formed in the mixed-clay minerals according to the XRD diffraction pattern. The corresponding XRD peaks are located at 20.827° and 26.624° (PDF#01-085-0457) for crystal planes of the quartz phase; 29.40°, 33.13°, 5.887°, 17.689°, 19.712°, 23.580°, and 34.743° (PDF#00-013-0135) for the montmorillonite phase; and 12.407°, 20.378°, 21.229°, 24.963°, and around 35°~39° (PDF#01-078-2110) for the kaolinite phase [32,33,34].

Figure 4 shows the FTIR spectra obtained for the three samples. For montmorillonite, the absorption peaks at 3692 cm^−1^ and 3617 cm^−1^ are metal hydroxide coordination hydroxyl -OH and interlayer water hydroxyl -OH stretching vibrations, respectively, the absorption peak at 3441 cm^−1^ is due to Si-OH, Al-OH, and intramolecular hydrogen bonding [35,36], and the absorption peak at 2364 cm^−1^ is the antisymmetric stretching vibration of CO_2_ molecules adsorbed in air. The absorption peaks at 1100–1000 cm^−1^ are the stretching vibration of Si-O, those at 918 cm^−1^ are the bending vibrations of Al-OH, those at 787 cm^−1^ are the stretching vibrations of Al-O, those at 691 cm^−1^ are the vertical vibrations of Al-OH, and those at 540 cm^−1^ are the stretching vibrations of Si-O-Al [36,37]. The absorption peak at 467 cm^−1^ is the bending vibration of Al-O and Si-O. Kaolinite is essentially the same as montmorillonite, and the absorption peak at 516 cm^−1^ should also be a stretching vibration of Si-O-Al [38,39].

Montmorillonite is a 2:1 aluminosilicate mineral consisting of two silica–oxygen flakes and one alumina flake combined into a wafer (layer) unit; these are then stacked on top of each other. Each layer has an oxygen ion group on both sides (on the silicon–oxygen sheet), so that no hydrogen bonds can be formed between the layers when they are stacked, but they are linked by “oxygen bridges”, which are weak and fragile, and the grains are smaller than those of kaolinite. The high-valence Al^3+^ is often replaced by the low-valence Mg^2+^ and Fe^2+^, and Si^4+^ by Al^3+^, resulting in montmorillonite having an excess of negative charge, and the weak linkage between layers in the structure makes montmorillonite strongly hygroscopic [40,41]. This is why its peak around 3600 cm^−1^ is stronger than the peak around 3400 cm^−1^ [42]. Kaolinite is (a 1:1 type aluminosilicate mineral) a lamellar lattice structure consisting of a silica–oxygen and a water–aluminum flake, connected by sharing the oxygen atoms at the top of the silica–oxygen flake. The wafers are strongly bonded by the formation of hydrogen bonds between them, making it difficult for water molecules and other ions to enter between the layers and form larger particles, making it less hygroscopic, bonding-capable, and malleable than montmorillonite [43]. This is why kaolinite has a larger peak at around 3400 cm^−1^ in its infrared spectrum.

### 3.2. Sensitivity of Humidity Sensor

By adjusting the relative humidity of the system environment, the deflections of MC-M, MC-K, and MC-MK sensors were examined in the relative humidity range of 10% RH to 80% RH. The sensitivity (*S_MC_*) of the MC humidity sensor at different relative humidity levels was calculated as:(2)$$S_{MC}=\frac{Z-Z_{0}}{∆RH}\times 100\%$$
where Z_0_ is the initial deflection of MC, Z is the deflection of MC at a different humidity, and ΔRH is the difference in relative humidity before and after the sensor undergoes deflection. The sensitivity response curves of the three humidity sensors at various humidity levels are depicted in Figure 5, and Figure 5a displays the fitted plots of the sensors’ deflections as a function of relative humidity; the inset section shows a schematic of the deflection change after the sensors adsorbed water. The linear coefficients of the MC-M, MC-MK, and MC-K sensors are 0.994, 0.946, and 0.991, respectively. The deflections of the MC-M, MC-K, and MC-MK sensors increased linearly with relative humidity, and a single clay exhibits good linearity above 99%. The dynamic real-time profiles are displayed in Figure 5b. For sensors under varying humidity environments, the deflection first rose quickly before gradually reaching a steady state. Sensors at baseline humidity showed deflections returning to the initial baseline value.

Among the three MC sensors (Figure 5), the MC-M sensor exhibited the highest moisture adsorption capability. Montmorillonite has a stronger adsorption for water than kaolinite, and the adsorption capacity of mixed clay is most affected by montmorillonite. The humidity exists in the form of water vapour during the measurement process, and the adsorption theory combined with the sensor’s moisture sensitivity mechanism is used to analyse the humidity sensitivity mechanism of the clay film MC humidity sensor, whose adsorption process is mainly physical adsorption. The hydrogen bonding in the clay makes the water molecules enter the interlayer of crystals, which expands the volume of the clay film.

The layered structure of kaolinite consists of alternating silica–oxygen tetrahedra and hydroxide ions. This structure gives kaolinite a large specific surface area and pore structure. Upon contact with water molecules due to hydrogen bonding, water molecules are more easily adsorbed onto the surface of kaolinite. When the humidity in the environment increases, water molecules are able to enter and be adsorbed into the pores of kaolinite, which are able to store a large number of water molecules. The adsorption of water molecules makes the spacing of the layer structure of kaolinite smaller, and the interaction between layers is enhanced, which makes the structure of kaolinite shrink or deform, so that the volume of kaolinite film changes. Compared to kaolinite, montmorillonite has larger interstices and pores in its layered structure, and these voids can hold more water molecules. In addition, montmorillonite has a strong ion-exchange capacity. When water molecules enter the interlayer voids of montmorillonite, some of the ions in the layered structure exchange ions with those in the montmorillonite, which increases the spacing between the montmorillonite layers and leads to swelling. This ion-exchange capacity promotes an increase in the amount of water absorbed by the montmorillonite.

The montmorillonite humidity sensor is approximately 27 times more sensitive to humidity than the kaolinite humidity sensor in the relative humidity range of 10% RH to 80% RH. However, as humidity increases, the MC-M sensor’s linear relationship changes slightly at relative humidity values from 63% RH to 80% RH, which may be explained by the presence of active adsorption sites for ambient water molecules in montmorillonite. In other words, high humidity levels resulted in elevated concentrations of cations between the montmorillonite layers, allowing water molecules to enter the crystalline layer under the influence of osmotic pressure. This formed a diffuse double-electric layer influenced by double-electric layer repulsion and increased the spacing between the crystalline layers. The MC then experienced an abrupt increase in the number of water molecules at the active point, leading to an enhanced mass of the montmorillonite film on its surface and an increase in the stress acting on it, thereby shifting the stresses above and below the MC as a function of the rise in humidity. This may deflect the linearity of the montmorillonite humidity sensor under 63% RH to 80% RH humidity conditions.

Overall, the MC-M exhibits superior sensitivity and linear correlation compared to the MC-K sensor. The sensitivities of the MC-M, MC-MK, and MC-K sensors under various humidity levels are compared in Table 1. The balanced sensitivity and linearity of the MC-MK sensor would make it more suitable for practical applications.

### 3.3. Humidity Hysteresis Properties

The humidity hysteresis is defined as the ratio of the maximum difference between the adsorption and desorption response curves to the relative humidity. For moisture adsorption, the sensors were first subjected to relative humidity environments changing from 10% RH to 80% RH by taking a test point every 5 min at humidity intervals of 17% RH to yield the deflection of the MC output of the sensor and the adsorption curves. Regulated humidity was then reduced from 80% RH to 10% RH, and the test curves representing the desorption were drawn. Figure 6a shows the dynamic hysteresis curves of the MC-M, MC-K, and MC-MK humidity sensors, while Figure 6b–d display the respective humidity hysteresis profiles. The moisture absorption and dehumidification curves of the sensors did not match perfectly. The respective humidity hysteresis values of the MC-M, MC-K, and MC-MK sensors were calculated as 5.83% RH, 0.31% RH, and 3.98% RH, respectively. Thus, the humidity lag of the MC-M sensor was longer than those of the MC-MK and MC-K sensors. This can be explained by the unexpandable mineral aspect of kaolinite, which leads to an adsorption process controlled by the surface hydration mechanism with no expansion of the crystal structure. The adsorption and desorption process consisted mostly of a reversible physical process, explaining why the MC-K hysteresis can hardly be noticeable. However, in montmorillonite, a swellable mineral, the adsorption process was controlled by the surface hydration mechanism. It included interlayer cation hydration, resulting in substantial hysteresis of the MC-M sensor.

### 3.4. Response and Recovery Time

The response time and recovery time curves of the MC-M, MC-K, and MC-MK humidity sensors under variable relative humidity levels are gathered in Figure 7a,b. The graphs show that the MC-M, MC-K, and MC-MK humidity sensors responded in about 52 s, 22 s, and 49 s, respectively. Note that the response time was measured from the initial state to the steady state as relative humidity increased from 10% RH to 80% RH, after which stabilization occurred. A subsequent decline in relative humidity from 80% RH to 10% RH resulted in good operation of the MC-M, MC-K, and MC-MK humidity sensors. After 97 s, 19 s, and 66 s, the MC-M, MC-K, and MC-MK sensors regained their baseline states, respectively.

To investigate the repeatability of the clay mineral humidity sensors throughout the complete wet range, the adsorption and resolution processes of all three humidity sensors were tested for successive cycles from 10% RH to 80% RH. As shown in Figure 8, all three humidity sensors revealed identical adsorption and desorption processes during the four cycles. At 80% RH, the response values looked nearly equal, and all responses were restored to their initial values after cycling. The response values at the 80% RH steady state were nearly identical. All sensors returned to their initial condition after cycling, demonstrating the high repeatability of the MC-M, MC-K, and MC-MK humidity sensors. The performance of the present sensor and reported work are summarized in Table 2 for comparison.

### 3.5. Stability

To evaluate the stability of the clay mineral films, the MC-M, MC-K, and MC-MK sensors were exposed to five humidity conditions by varying the humidity (10%, 27%, 45%, 63%, and 80%) of the chamber at five time points after certain numbers of days (0, 5, 10, 20, and 30 days, respectively), followed by measurement of the sensor deflection. The total time for stability testing was 30 days, and responses of sensors were tested at five time points: 0, 5, 10, 20, and 30 days, respectively. The results are shown in Figure 9. As the long-term stability curves (Figure 9) show, the MC-K humidity sensor showed slight fluctuations after 15 days. The MC-M sensor had better stability than the MC-K sensor, while the MC-MK sensor was somewhere in between. Overall, the sensors exhibited good long-term stability.

## 4. Conclusions

Lightweight, inexpensive, and flexible MEMS humidity sensors were developed using clay minerals. Properties such as deflection, sensitivity, wet hysteresis, repeatability, and stability of the montmorillonite-, kaolinite-, and mixed-minerals-coated MC sensors for humidity were all evaluated. All MCs showed good linearity, repeatability, and long-term stability over 30 days. The sensitivity of the montmorillonite humidity sensor was higher than that of the kaolinite humidity sensor, with up to 140.9 nm/%RH. However, the MC-M had poor linearity in the humidity range from 63% RH to 80% RH. The kaolinite humidity sensor displayed the lowest moisture hysteresis of 0.31%, with twice the response and five times the recovery speed of montmorillonite humidity sensors. By comparison, the performance of the montmorillonite/kaolinite mixed-dispersant humidity sensor was somewhere in between, making it more suitable for practical applications. In sum, the proposed clay-based MEMS humidity sensors look to have potential for preparing high-performance humidity sensors, offering a fresh concept for manufacturing new humidity sensors.

## Figures and Tables

**Figure 1 sensors-23-06962-f001:**
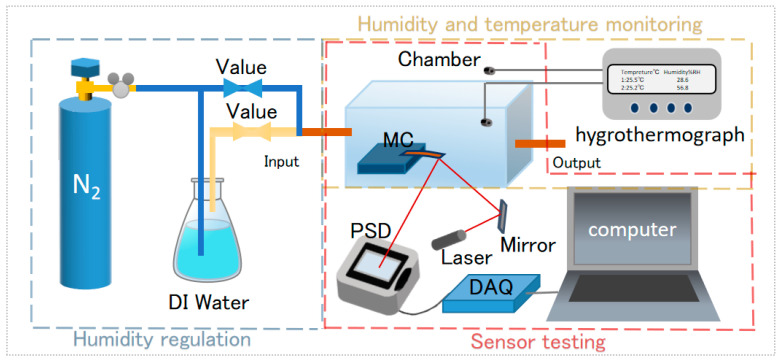
The schematic block diagram of the micro-cantilever humidity sensor setup.

**Figure 2 sensors-23-06962-f002:**
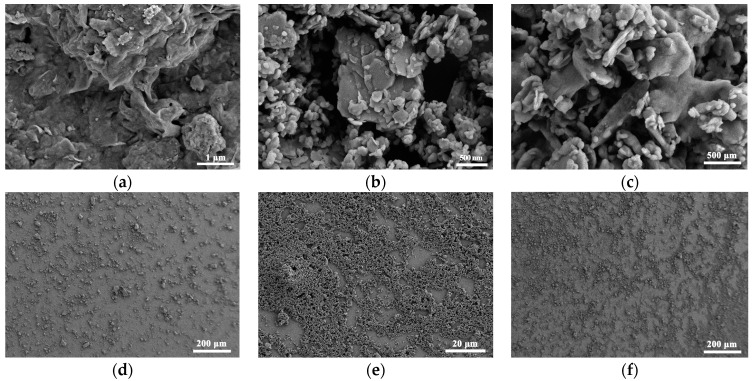
SEM images of (**a**) montmorillonite film, (**b**) kaolinite film, and (**c**) mixed-clay film. Macroscopic images of (**d**) montmorillonite film, (**e**) kaolinite film, and (**f**) mixed-clay film.

**Figure 3 sensors-23-06962-f003:**
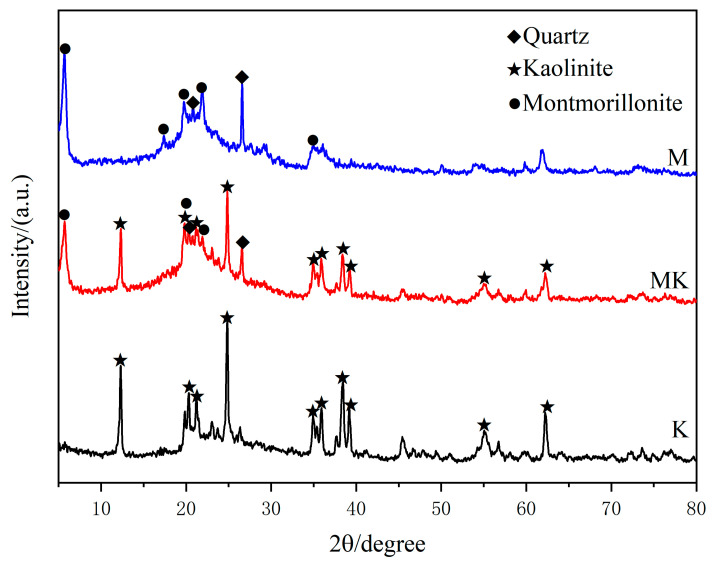
XRD patterns of montmorillonite, kaolinite, and mixed-clay minerals.

**Figure 4 sensors-23-06962-f004:**
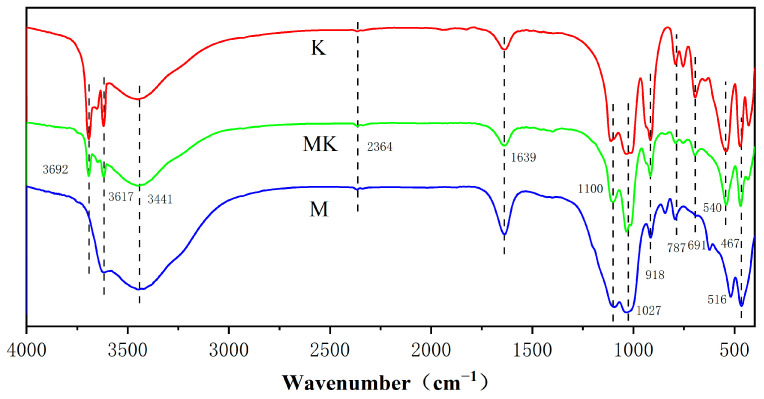
FTIR spectra of montmorillonite film, kaolinite, and mixed-clay minerals.

**Figure 5 sensors-23-06962-f005:**
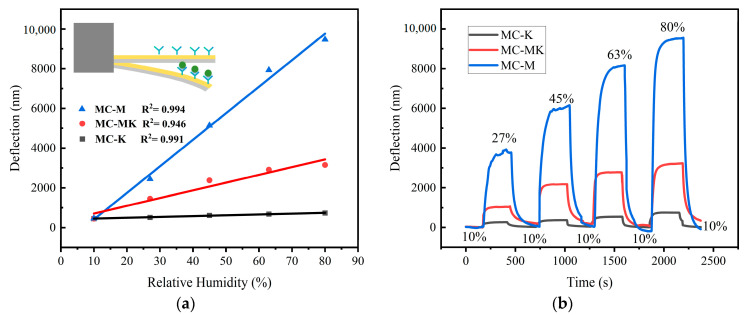
Response curves of humidity sensors: (**a**) humidity–deflection fitting curves and (**b**) real-time dynamic curves.

**Figure 6 sensors-23-06962-f006:**
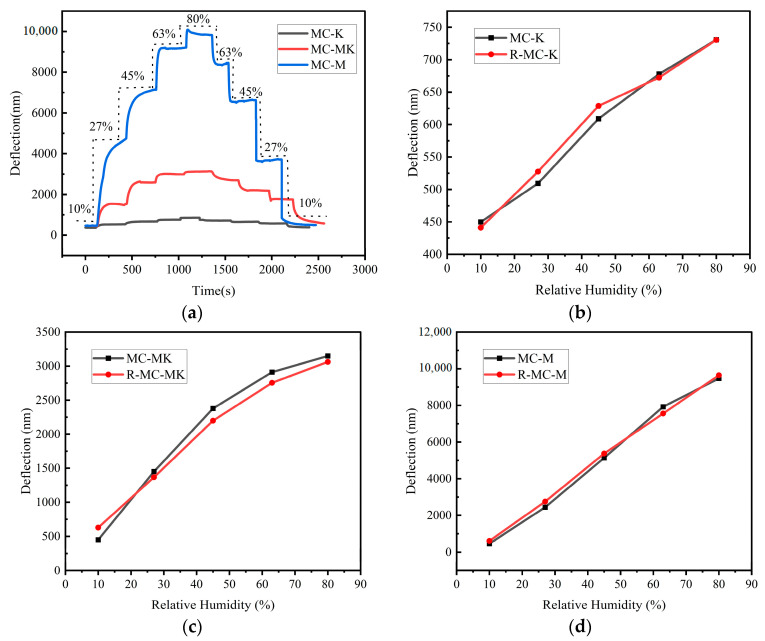
Wet hysteresis curves of micro-cantilever humidity sensors: (**a**) wet hysteresis dynamic curves, (**b**) MC-K, (**c**) MC-MK, and (**d**) MC-M.

**Figure 7 sensors-23-06962-f007:**
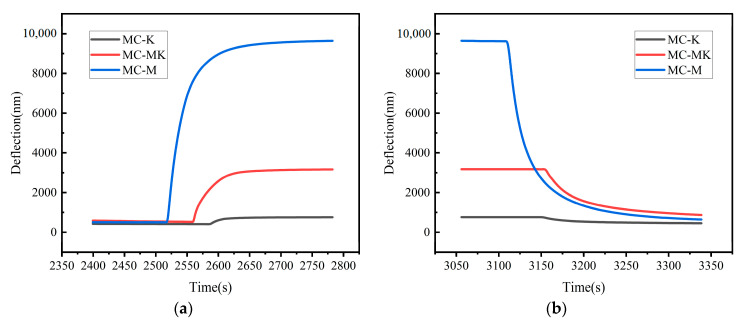
Response (**a**) and recovery curves (**b**) of the humidity sensors.

**Figure 8 sensors-23-06962-f008:**
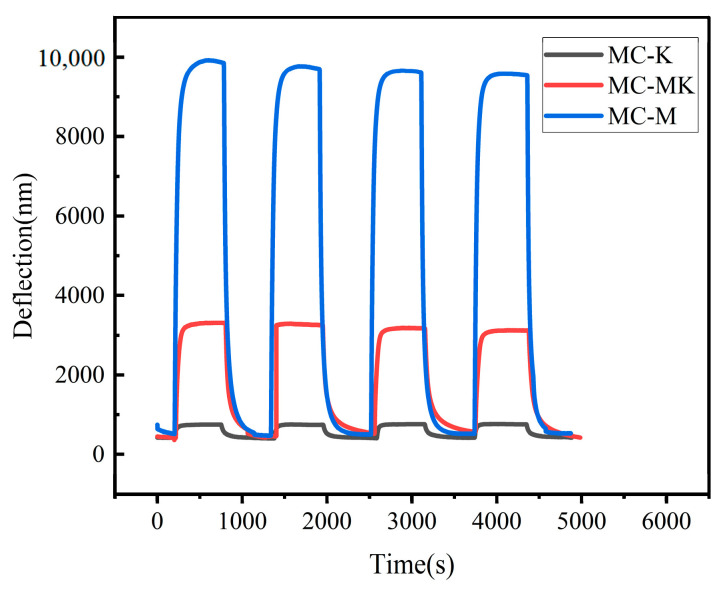
Repeated curves of humidity sensors during cycling.

**Figure 9 sensors-23-06962-f009:**
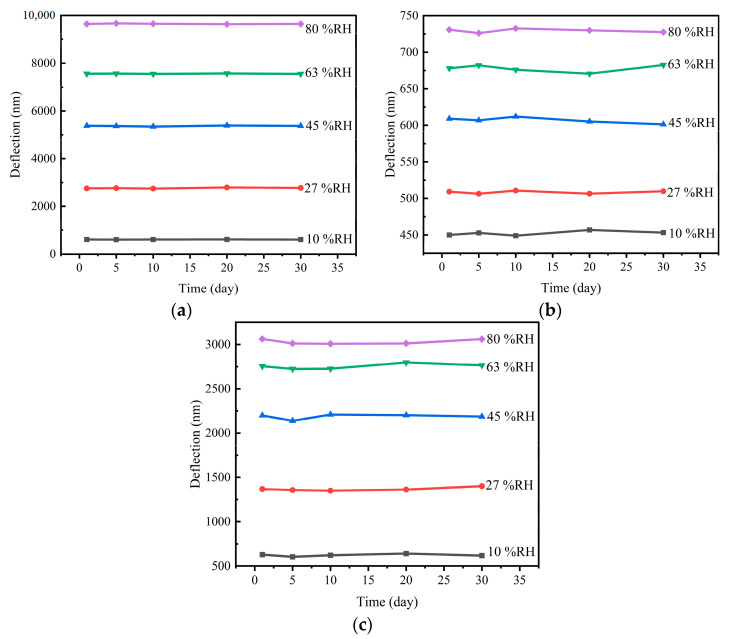
Stability curves of micro-cantilever humidity sensors: (**a**) MC-M, (**b**) MC-K, and (**c**) MC-MK.

**Table 1 sensors-23-06962-t001:** Sensitivities of MC-M-25, MC-M-50, and MC-M-100 sensors under different humidity levels (nm/%RH).

Sensitivity/S	10–27 (nm/%RH)	10–45 (nm/%RH)	10–63 (nm/%RH)	10–80 (nm/%RH)
MC-M	116.7	133.6	140.9	128.7
MC-K	3.49	4.54	4.3	4.01
MC-MK	58.9	55.1	46.4	38.5

**Table 2 sensors-23-06962-t002:** Comparison between MC humidity sensor in this work and reported work.

Reference	Sensitive Material	Sensing Range (%RH)	Sensitivity	Response and Recovery Time (s)
This paper	Montmorillonite	10–80	128.7 nm/%RH	52/97
[44]	polyaniline	20–65	121.4 nm/%RH	-
[45]	Complementary metal oxide semiconductor	20–80	7 mV/%RH	-
[46]	Molybdenum disulfide	10–90	778 Hz/%RH	0.6/8
[47]	Graphene oxide	10–90	84.41 Hz/%RH	10/10

## Data Availability

Not applicable.

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
