# Peer review of "Fabrication of Micro-Cantilever Sensor Based on Clay Minerals for Humidity Detection"

_sensors, 2023, doi:10.3390/s23156962_

Round 1

Reviewer 1 Report

1.please provide physical images of the device and sensor testing systems, and give detailed description for testing schematic.

2. what is the effect of temperature on the sensor's humidity sensing performance?

3.what are the influences of the other ratios of composite films, the author only considers the ratio of 1:1. 

4.The humidity sensing mechanisms should be further investigated and analyzed.

Reviewer 2 Report

The authors have proposed a flexible MEMS humidity sensor developed using clay minerals.

The method of data extraction and preparation  is well- described  and the article is clear . 

Author Response

Dear reviewer,

Thanks you for offering us an opportunity to submit manuscript. We sincerely appreciated for your comments and understanding.

 We tried our best to improve the manuscript and made some changes in the manuscript. These changes will not influence the content and framework of the paper. And here we did not list the changes but marked in red in revised paper.

Thank you very much for your attention and time. Look forward to hearing from you.

Reviewer 3 Report

Dear Authors,

Congratulations for your work on designing MC sensor for humidity.

Before publication you should address some issues with respect to the experimental setup.

- how were the layers of clay minerals prepared? What technique did you used?

- the entire dispersion of 4 mL was casted on the MC array? Are there any thickness measurements for the sensing layer?

- You present exactly 5 points of humidity (10%, 27 %, 45 %, 63 % and 80 %) on your graphs, obtained in a repetitive way. How was varied humidity in the testing chamber? How did you manage to obtain the same humidity values for several cycle of sensor functioning? You present fixed values of humidity measurement not only in several cycles of sensor functioning, but also in 5 different days in the 30 days of the stability test.

- please give a detailed description of the testing chamber

-figure 1 has many abbreviations unexplained

- in the characterization section you have to present solid references for all your allegations (XRD diffraction peaks, IR absorption maxima assignements).

Reviewer 4 Report

Before ready for publications, I have several concerns below:

1. XRD pattern does not have reference for MC and K, please include the database reference to in the XRD as well to ensure the purity of the as-received products.

2. Moreover, the assignment of the functional groups in the FTIR did not refer to any reference, which are not appropriate when analyzing the FTIR results. It also includes the non-superscript format for representing the wavenumber.

3. What is going to happen if the developed sensors are exposed to RH higher than 80%?

4. In Figure 5(a), Figure 6(b),(c), and (d), authors should include the fitting of the deflection vs RH to see how the strong relation between the RH and deflection. It is important for practical relevance. Or at least a graph is needed to show the sensitivity vs relative humidity.

5. How do authors define Sensitivity? How to calculate it? It did not appear in the manuscript.

6. Is there any structure alteration after stability test, e.g. from the FTIR result?

7. Finally, a table representing a comparison between this research and other previous reports is desirable to showcase how good the developed sensors is.

Round 2

Reviewer 1 Report

This manuscript has been revised, and my question has been addressed carefully, so i recommend to publish in its current form.

Reviewer 4 Report

The authors have addressed the concerns raised by reviewer. The manuscript is now can be accepted in current form.